# Assessment of Household Food Security in Fish Farming Communities in Ghana

**Akua S. Akuffo \* and Kwamena K. Quagrainie** 

Department of Agricultural Economics, Purdue University, 403 W. State Street, West Lafayette, IN 47907, USA; kquagrai@purdue.edu
**\*** Correspondence: asakuffo@gmail.com

**Abstract:** The Government of Ghana and international NGOs have been encouraging the adoption of fish farming to alleviate poverty and food insecurity through training workshops, financial contributions and creation of a fisheries ministry. Nevertheless, there is no study on how these efforts have influenced the household's welfare, particularly their nutritional quality. Based on this, our objective is to identify the ways through which fish farming impacts the household's nutritional quality. We hypothesize that engaging in fish farming will increase steady income flow and access to fish for the household's direct consumption. We adopted the Propensity Score Matching (PSM) approach in a logit framework to achieve this objective and address the endogeneity from the bias of self -selection by creating a statistically similar-looking control group. The results suggest that fish farming households have higher nutritional quality and frequency of food consumed than the non-fish farming households through direct consumption. The probability of adopting fish farming increases with wealth, location, ecological zone and household size but decreases with household income per capita. The average effect of adopting fish farming on household nutritional quality is 15.5 Food Consumption Score points. Policies that encourage women to engage in not only fish processing, but production as well are advised.

**Keywords:** food security; fish farming; propensity score matching; food consumption score (FCS), Ghana

## 1. Introduction

Fish farming is becoming very popular in developing countries because of its ability to improve the welfare of particularly less wealthy and landless-food insecure households through employment, income generation and nutrition from direct consumption [1]. In Africa and Asia, several developmental interventions related to fish consumption, aquaculture, and capture fisheries have aimed at improving the nutritional status of households through direct dietary intake, production and increase in household income [1]. In addition to the provision of food, fish farming has the potential to contribute to the Millennium Development Goals (MDGs) of reducing poverty and halving hunger through creation of employment, community development [2] and consequently has been adopted by many Asian and African national governments.

In 2013, the Ministry of Fisheries and Aquaculture Development was created in Ghana out of the Ministry of Food and Agriculture to give more emphasis and support to the industry. The ministry, with support from the Food and Agriculture Organization (FAO), developed a Ghana National Aquaculture Development Plan (GNADP) to increase profitability and production up to 100,000 metric tons by 2016 [3]. The GNADP also aimed at improving the capacity for fish farmers through training in best management practices and feed formulation [4]. Similar capacity training was also conducted by the United States Agency for International Development (USAID), UK's Department

for International Development (DFID), and the New Partnership for Africa's Development (NEPAD). The World Bank, in collaboration with the Government of Ghana (GoG), also developed the Ghana Fisheries and Aquaculture Development Plan (GFADP) [5]. The GFADP serves as a roadmap by which aquaculture will contribute to poverty alleviation, food and nutritional security, employment generation, increased income and economic development as part of the government's efforts to reduce poverty under the Ghana Poverty Reduction Strategy II (GPRS II) [4] and the Millennium Development Goals. The Ghana Association of Women Entrepreneurs (GAWE) and Rural Wealth (RW) are among the few local non-governmental organizations that are actively engaged in aquaculture projects [6].

There are two main aquaculture production systems in Ghana-ponds and cages. Pond aquaculture constitutes the main production system and is adopted by small to medium-scale farms for the production of tilapia (Oreochromis niloticus) and catfish (Clarias gariepinus) in a polyculture semi-intensive system. Large-scale enterprises using pond systems adopt monoculture in intensive tilapia production. The cage culture sector also comprises small, medium and large-scale commercial enterprises engaged in intensive tilapia production in natural water bodies, such as lakes, reservoirs and rivers. One of the major constraints to aquaculture production is economic access to commercially formulated feed. Most of the available commercially formulated feed on the market are imported, and relatively expensive. Feed generally accounts for between 40–60 percent of production costs. Consequently, small to medium scale producers tend to formulate feed on the farm from a wide range of local feed ingredients including agricultural by-products such as maize bran, wheat bran, groundnut husks, various leaves, vegetables, etc. Producers may use the ingredients directly, make simple mixtures of ingredients, or compound ingredients, but these are generally of poor quality. Some producers use commercially formulated feed and supplement with farm-made feeds.

The level of production from aquaculture in Ghana increased from approximately 19,092 metric tons in 2011 to 44,515 metric tons in 2015 [7]. From 2009 to 2012, cage aquaculture production experienced an increase from 4912 metric tons to 24,249 metric tons [6,8]. The growth in production is mainly market-driven, with diversified and increased sales of aquaculture products. Economically, the expanded market opportunities for aquaculture small businesses and fish farmers ultimately lead to farm sustainability and increased profitability for businesses and farmers. There are also linkages to economic development in communities, which bring improvements in the rural agricultural economy. In addition, the growth of the aquaculture sub-sector and accompanying expanded activities have social implications including provision of employment and other social benefits to communities, especially rural communities along the Volta lake. However, there have been growing concerns about the sustainability of cage aquaculture in the Volta lake in Ghana because of the potential effects on the lake including biodiversity, escapes and effluent discharges [6,9]. However, given adequate regulatory infrastructure, aquaculture in Ghana can develop towards a sustainable alternative solution for the supply of seafood due to the decline in wild capture fisheries.

The per capita consumption of fish for the average Ghanaian is about 25 kg per annum, making Ghana one of the highest fish consumer in Sub-Saharan Africa. Fish represents about 60% of the animal protein consumed in Ghanaian homes [6]. The commonest types of fish consumed is tilapia and cat fish which are the most commonly farmed fish as well [10]. The empirical question that needs to be answered is whether fish farming contributes to a household's food security/nutritional quality in Ghana. If it does, in what ways? A country's socioeconomic development depends on the welfare of its citizens, particularly their health.

The food insecurity situation in Ghana is more about access and stability than availability. Food insecurity is a national issue in Ghana due to widespread poverty. The problem exists in both rural and urban areas with the rural areas being the most affected. Ghana's food insecurity is heightened by irregularities in the seasons and production. The latter is highly dependent on rainfall, high food prices and low incomes at the household level [10]. The most food insecure regions in Ghana are the three Northern regions and the least food insecure areas are Greater Accra and the Western regions [11].

The concept of food security is measured in so many ways due to lack of proper assessment of the different aspects of it [12]. In estimating the impact of technology adoption on food security, for example, proxies for food security include the use of household income and expenditure; others have also tried to use production measures [13]. Synthetic poverty indices have been used in some studies [14]. All these measures conclude that the adoption of an improved practice has a positive impact on the welfare of Sub-Saharan African (SSA) countries and contributes to the reduction of poverty. These measures, however, have limitations, mainly on how much impact can be captured with money and production [12]. Other indicators developed and validated by the World Food Program (WFP) and other organizations include Coping Strategies Index (CSI), Reduced Coping Strategies Index (rCSI), Household Food Insecurity and Access Scale (HFIAS), the Household Hunger Scale (HHS), Food Consumption Score (FCS), Household Dietary Diversity Scale (HDDS) and a self-assessed measure of food security (SAFS) [15].

Several factors that are critical to the nutritional decisions made by the household are the focus in studies of household nutrition. The common factors identified in the literature are income, tastes, education, family size and composition, and market price [16]. In Bangladesh, it was observed that even though fish is quite expensive, consumption in small quantities makes a significant difference in contributing to the nutritional quality of the diets of poor people [17]. In the central region of Malawi, a study of fish farming and non-fish farming households over a period of four weeks revealed no significant differences between households in terms of nutritional status. Fish farming households cultured the fish mainly for selling purposes and not for consumption; fish consumption by producing households was very low [18].

A study was carried out to evaluate the influence of food insecurity on the malnutrition of children. Using a sample of children aged 6–36 months residing in both rural and urban areas of Tamale in northern Ghana, access of households to food was measured using the Household Food Insecurity and Access Scale (HFIAS), Food Consumption Score (FCS), and Household Dietary Diversity Score (HDDS) [19]. The determinants that influenced malnutrition of children included wealth index, body mass index of the mother, mother's educational level, the area of residence and access to portable water. A study in Ghana revealed that even though poorer households had successfully adopted aquaculture, the impacts could not be determined from their livelihoods compared to non-poor fish farming households [20]. The author explained that the adoption of aquaculture was dependent on household characteristics and the level of knowledge about aquaculture and concluded that the level of impact is largely dependent on the socio-economic status of the household as well as institutional and infrastructural resources available to them.

This paper examined the impact of fish farming within the framework of an agricultural innovation since it involves the use of technology, which deviates from the traditional farming activities. The impact of a technology intervention on income, expenditure and other components of food security may be positive or negative for the household. We use data on all ten regions of Ghana, but most fish farmers are located along the Volta Lake in the Eastern, Brong Ahafo and Volta regions, making the decision to engage not random.

Even though fish farming has become an integral part of the efforts of the Ghanaian Government to reduce poverty and improve food security, there is no study on the assessment of how participating in fish farming impact the nutritional quality of farming households. The main objective is to identify the direction of impact and the pathway(s) of fish farming on household nutritional quality. We hypothesize that fish farming households have more diversified diets than non-fish farming households. The assumption is that engaging in fish farming will increase steady income flow and access to fish for household's consumption. Households will further be able to purchase and consume more diverse and nutritious food items, particularly vegetables, meat, dairy and fresh fruits.

The measure of food security is proxied by the World Food Program's Food Consumption Score (FCS). The correlation between fish farming and food security is estimated using the Average Treatment on the Treated (ATT) under the Propensity Score Matching (PSM) framework. We address three

main gaps in the literature on the impact of fish farming on household welfare. First, most of the impact studies have focused mainly on agricultural households without separating them into different sectors including fisheries and aquaculture. Secondly, the household welfare measures have focused on poverty alleviation and income growth and very rarely on nutritional improvements. Lastly, we address the lack of assessment of the counterfactual situation if the treatment had not been received, i.e., if fish farming had not been adopted. We use a non-parametric matching approach, the Propensity Score Matching (PSM) to address this issue.

## 2. Materials and Method

The decision for any individual or household to take up fish farming as an occupation is in most cases unknown to the researcher. Consequently, the attempt to assess the magnitude or ways in which fish farming contributes to a household's nutritional quality is not straightforward, especially in the absence of pre-participation data. The literature refers to this as self-selection. Regression estimates will either have an upward or downward bias. Alternative ways that address the bias include the Heckman Two-Step Approach, one of the popular methods that uses the Inverse-Mills ratio as an explanatory variable in the outcome equation. The shortcoming of the approach is the normal distribution assumption made about the unobserved variables. Another popular alternative is the Instrumental Variable (IV) approach because it deals with both hidden and overt biases. However, it is difficult to find a variable, which is correlated with the decision variable but uncorrelated with the outcome variable, which makes the approach unattractive. Several instruments were identified and tested (Omitted variable test, Oster's test for omitted variables, Durbin-Wu-Hausman test) in this study but none was found to be suitable.

*Propensity Score Matching (PSM)*

Propensity Score Matching (PSM) is a quasi-experimental approach. PSM reduces the estimation bias in measuring the impact of a treatment with observational data [21]. The non-random assignment of the treatment and control groups introduces a selection bias. PSM involves mainly two stages; estimating the decision to adopt fish farming in a logit regression with fish farming as a function of household observable characteristics. The second stage is the determination of the impact of the adoption decision on household nutritional quality, the outcome variable using the average treatment effect. The impact of the adoption decision on the outcome variable is estimated by calculating the net impact of adoption on the household's nutritional quality [8]. The following segments give details about the two stages of the PSM. The basic set up for PSM is

$$Y_1 = \beta_1 X + \varepsilon_1$$
$$Y_0 = \beta_0 X + \varepsilon_0 \tag{1}$$

where $Y_1$ is the outcome variable for the treated group, that is, fish farming households; $Y_0$ is the outcome variable for the control group, that is, non-fish farming households; X is the vector of observed characteristics for both control and treated groups, and $\varepsilon_1$ and $\varepsilon_0$ represent the error terms assumed to be exogenous of the vector of observed covariates. The ideal case for such an impact study is to make a comparison for the same individual with and without the treatment. However, it is not possible in this study due to the absence of a pre-participation data of the individuals. One way to deal with the absence of pre-participation data is to do a counterfactual analysis [22]. The practical approach is to have an "observed" outcome $Y_1$ if an individual is a fish farmer and the counterfactual or control $Y_0$ outcome that would have happened if the individual were not a fish farmer. The difference between what happened and what would have happened is expressed as:

$$\text{net impact of adoption} = Y_1 - Y_0. \tag{2}$$

Our interest here is the correlation between participation in fish farming and the household's dietary diversity. Following Rubin and Rosenbaum, we estimate the Average Treatment on the Treated (ATT) using two matching algorithms; nearest neighbor (NNM) and kernel-based matching [23–25]. After the matching, the bias ratio of the covariates is tested using a Likelihood ratio test and pseuso-$R^2$. The pseudo-$R^2$ should be low, and we should reject the Likelihood ratio test for joint significance. The Rosenbaum Sensitivity analysis [26] tests whether unobserved covariates are influencing our estimates as this will affect our final inferences (Details shown in Supplementary Materials).

## 3. Data

Data for the analysis were obtained from the 2013 Ghana Living Standards Survey (GLSS6) [27]. The survey solicited information on a range of factors including demographic, socioeconomic, individual, household characteristics and health issues. The data has information on a total of 16,772 households from all ten regions of Ghana, but we used approximately 4011 household information that included 144 fish-farming household. The sample size was determined after using influence diagnostics (Influential diagnostics will generally produce large studentized residuals (rstudent) for outliers. Using rstudent thresholds, observations were dropped if their rstudent values were outside the range of $2 \geq r \leq -2$) to identify observations that influence our variance and everything else.

*Outcome and Treatment Variables*

*Food Consumption Score (FCS):* The FCS is used as a 'validated proxy measure' for food security [14]. FCS was developed with funding from USAID under the Food and Nutrition Technical Assistance (FANTA) project and the World Food Program (WFP) promotes its use. It is a weighted measure of dietary diversity and constructed as:

$$FCS = \sum y_i\, f_i \tag{3}$$

where $y_i$ is the different food groups and $i$ represents cereals, roots & tubers, vegetables, fruits, meat/poultry, eggs, seafood/fish, milk, legumes, sugar, oil/fat and condiments; $f_i$ is the consumption frequency of that food group over the past week or month (Table 1). We hypothesize that fish farming households have higher dietary diversity and food security than the non-fish farming households. The assumption is that engaging in fish farming is expected to have a positive and direct impact on household income. Households will then be able to purchase and consume more diverse and nutritious food items, particularly vegetables, meat, dairy and fresh fruits.

**Table 1.** World Food Program (WFP) food groups and weights used in calculating food consumption score.

| Food Items | Food Groups | Weights |
|---|---|---|
| Maize, maize porridge, rice, sorghum, millet, pasta, bread, other cereals, Cassava, potatoes & sweet potatoes | Cereals & Tubers | 2 |
| Beans, peas, groundnuts, cashew nuts & other nuts | Pulses | 3 |
| Vegetables, leave & fruits | Vegetables &fruits | 1 |
| Red meat, poultry, eggs, fish | Meat &fish | 4 |
| Milk, yoghurt & other dairy products | Milk | 4 |
| Sugar & sugar products | Sugar | 0.5 |
| Oils, fat & butter | Oil | 0.5 |
| Condiments | Condiments | 0 |

Data at the household level provides information on the characteristics and location of the household. This study uses data on the following demographics of the household head: years of education, marital status, sex, employment status and age. Additional information on the household's location and economic status are proxied by ecological zone, wealth index, household size and monthly income in Ghana cedis (GHS).

In identifying the food security status of a household, the thresholds from the WFP are used. FCS has two different range values depending on the frequency of sugar and oil consumed by the

household. Households with FCS of 0–21 are categorized as having poor food consumption, those within 21.5–35 are classified as borderline food consumption, and those above 35 are categorized as having acceptable food consumption. According to the WFP, the FCS value can be adjusted in the presence of evidence [28]. Different countries have different thresholds based on their situation. For example, in Laos and Haiti, the threshold levels are much higher for daily edible oil and sugar consumers. The situation in Laos and Haiti is like our sample therefore; we adopted the threshold levels used in Laos and Haiti.

*Fish farming*: This is a binary variable representing whether the household is engaged in fish farming. It is the dependent variable in the logit regression in PSM. It is assumed that households engaged in fish farming consume the fish they harvest from their own ponds and obtain income from selling some of the fish. The variable is expected to have a positive correlation with food security.

## 4. Results and Discussion

### 4.1. Summary Statistics

The average FCS is 57.5 units with fish farming households on average having approximately 68.7 while non-fish farming households have a consumption score of approximately 57.3 (Table 2).

**Table 2.** Summary statistics.

|  | Unit | Pooled | Sd | FFHH Mean | nFFHH Mean | FFHH Sd | nFFHH Sd | T-Test (*p* Values) |
|---|---|---|---|---|---|---|---|---|
| FCS |  | 57.49 | 13.67 | 68.74 | 57.33 | 14.98 | 13.38 | 0.00 |
| Wealthindex_sqr |  | 1.56 | 2.66 | 1.58 | 1.56 | 2.78 | 2.65 | 0.01 |
| Wealth index |  | −0.19 | 1.26 | −0.21 | −0.19 | 1.47 | 1.25 | 0.00 |
| Education | Years | 10.28 | 6.12 | 7.00 | 10.33 | 6.62 | 6.09 | 0.00 |
| Age | Years | 45.16 | 12.08 | 44.79 | 45.16 | 12.32 | 12.07 | 0.18 |
| Peri-urban |  | 0.05 | 0.27 | 0.03 | 0.05 | 0.41 | 0.26 | 0.00 |
| Marital status |  | 0.77 | 0.39 | 0.93 | 0.77 | 0.26 | 0.39 | 0.00 |
| Employed |  | 0.99 | 0.10 | 1.00 | 0.99 | 0.17 | 0.10 | 0.04 |
| Male |  | 0.81 | 0.36 | 1.00 | 0.81 | 0.27 | 0.37 | 0.01 |
| Ecology |  | 2.27 | 0.64 | 2.65 | 2.26 | 0.63 | 0.64 | 0.00 |
| HHinc_cap | USD | 248.37 | 545.82 | 34.35 | 251.40 | 124.56 | 554.16 | 0.00 |
| HHsize |  | 5.48 | 3.26 | 8.39 | 5.44 | 4.87 | 3.16 | 0.00 |
| Techsupport |  | 0.49 | 0.50 | 0.49 | 0.49 | 0.49 | 0.50 | 0.00 |

HHinc_cap is household income per capita, HHsize stands for household size., Techsupport represents technical support, Wealthindex_sqr is the wealth index variable that is squared to capture any non-linear relations between wealth index and the dependent variable. FCS: food consumption score, FFHH: Fish farming household, nFFHH: non-fish farming households.

Fish farming households also have higher absolute wealth index values than non-fish farming households, with 0.21 and 0.19, respectively (Table 3). Only 4.7 percent of the sample is in the peri-urban area, with fish farming households making up 3.2 percent and non-fish farming households, 4.8 percent (Table 3). The average household size for a fish farming household is 8 compared to 5 for the non-fish farming household.

The control group is mostly agricultural households. On the average, approximately 49 percent of our sample receives some type of extension assistance. The average income per capita for fish farming households is low (34.35 GHS) compared to non-fish farming households (251.40 GHS). The same pattern is observed in the average years of education, 7 years for fish farming household heads and 10 years for non-fish farming household heads. The fish farming households differed significantly in most of the household characteristics from non-fish farming households except for age, which was highly insignificant (Table 3).

**Table 3.** Results showing Factors that Influence Fish Farming Adoption Decision.

| Variables | Coefficient (SE) | Average Marginal Effects | Odds Ratio |
|---|---|---|---|
| Wealth index squared | 0.382 | 0.012 | 1.465 |
| | (0.05) *** | (0.00) ** | |
| Wealth index | 0.802 | 0.029 | 2.230 |
| | (0.107) | (0.00) *** | |
| Education | −0.024 | −0.001 | 0.976 |
| | (0.02) | (0.00) | |
| Age | 0.001 | 0.000 | 1.002 |
| | (0.01) | (0.00) | |
| Peri-urban | 0.953 | 0.030 | 2.593 |
| | (0.23) *** | (0.01) *** | |
| Married | 0.074 | 0.002 | 1.077 |
| | (0.41) | (0.01) | |
| Employed | −0.214 | −0.007 | 0.807 |
| | (0.59) | (0.02) | |
| Sex | 0.608 | 0.019 | 1.836 |
| | (0.39) | (0.01) | |
| Ecology | 0.319 | 0.010 | 1.376 |
| | (0.17) * | (0.01) * | |
| Income/capita | −0.003 | −0.000 | 0.997 |
| | (0.00) *** | (0.00) *** | |
| Household size | 0.099 | 0.003 | 1.104 |
| | (0.02) *** | (0.00) *** | |
| Constant | −5.431 | | 0.004 |
| | (0.90) *** | | |

* $p < 0.1$, ** $p < 0.05$, *** $p < 0.01$.

### 4.2. Factors Influencing the Adoption of Fish Farming at the Household Level

Table 3 shows the coefficients, average marginal effects and the odds ratio with respect to the outcome variable, FCS. Apart from education, sex, age, employment status and marital status of the household head that were not significant, the rest of the covariates were significant.

The marginal effect for peri-urban is positive and significant at the 1 percent level (Table 3). The implication here is that a household's adoption decision increases by 3.0 percent with an increase in proximity to, or residence in, a peri-urban area. Most of the fish farms in Ghana are in peri-urban areas. These areas tend to have a land size suitable for agriculture, amenities to enhance its success such as nearness to market centers to take advantage of competitive prices and low transportation costs, particularly for smallholder farmers [20,29].

Residents of peri-urban localities also have a higher probability of finding other off-farm income-generating opportunities [30], access to an energy source (electricity) and increased access to technical support as the extension officers are hardly ever willing to travel to rural areas. Most of these areas are commonly located in the Ashanti, Brong Ahafo, Western, Greater Accra and Eastern regions of Ghana.

Most studies on the factors that influence the decision to adopt an agricultural-related technology use a sample in the rural areas, e.g., [25,31–34]. Our results cannot be directly compared to these studies because they focused on rural and peri-urban household market participation and the factors that influence them. However, some studies have concluded that households in the peri-urban areas had greater access to information as well as amenities making the decision to participate much easier compared to those in the rural areas.

The total average marginal effect of wealth on fish adoption is 4.1 percentage points (1.2 + 2.9). This average implies that on the margin, a 1 percent increase in wealth increases the probability to adopt fish farming by 4.1 percentage points at the 1 percent level. The probability to adopt fish farming increases the wealthier a household gets. Studies by [34] and [35] all found similar results, while [36] reported a negative but significant coefficient in Niger.

The wealth index, a proxy measure of the long-term economic status of the household is positive as we hypothesized. Wealth is a strong indicator of the economic situation of the household and a good predictor of the household's attitude toward technology adoption. A wealthy household can assume risks associated with technology, has greater access to resources, credit and diverse income sources, which increases their probability of adopting new technology, especially those that are capital intensive.

The ecological location of the household head increases the possibility of adoption at the 5 percent level (Table 3). The marginal effect of the ecological zone means that the probability of the household adopting fish farming increases by 1.0 percentage point with a 1 percent move toward the inland regions. Similar results are reported by [23,37]. However, other studies such as [38] and [12] found contradictory results. Ghana has three main ecological zones, namely the savannah found in the Brong Ahafo, Northern, Upper East and West regions, the forest found in the Western, Central, Volta, Eastern, Ashanti and parts of the Brong Ahafo regions and coastal regions found in the southern parts of Greater Accra, Central, Western and Volta regions. Those in the coastal regions are mainly engaged in marine fishing due to their proximity to the sea.

The income effect of adopting fish farming on household nutritional quality was not as strong as we expected, even though it was statistically significant in the opposite direction. The negative marginal effect implies that, on the margin, as per capita income declines, we expect aquaculture adoption to increase. Households involved in fish farming are generally in the lower-income group and fish farming tends to be a secondary or supplementary occupation. This appears to confirm that the household wealth index measure based on household assets may be a better proxy indicator for economic status than income.

The results suggest that fish farming will positively influence the nutritional quality of households in the forest and savannah regions. The forest regions have long rainy seasons and water resources to support fish farming. Due to the dry nature of the savannah ecological zone, agriculture is dependent on more water reservoirs for irrigation. There are existing irrigation schemes around the country, including Vea, Tono, in the Upper East and Golinga, Ligba and Bontanga and Northern regions, which support rice and vegetable production [39]. These schemes can be good sources of water for cage culture, which requires minimum initial investment compared to pond aquaculture that requires materials and labor for construction and other pond management activities.

From Table 3, the size of the household also positively influences the probability of adoption at the 1 percent level. The probability of adopting fish farming increases by 0.3 percentage points on the margin when the household size grows by one person. Similar results were also recorded by [40,41] in their studies. However, contrary results were identified by [42–44]. Labor is a costly input in agriculture in Ghana making family members the main source of labor for most subsistence fish farming households.

*4.3. Impact of Fish Farming on Household Nutritional Quality*

In order to estimate the impact fish farming has on household nutritional quality, the average treatment effect on the treated (ATT) was calculated after matching, and the results are shown in Table 4. All matching algorithms show similar results—on average, adopting fish farming significantly increases the nutritional quality of households as measured by the food consumption score. The average improvement in nutritional quality is between 13.9 and 15.5 points. This increase in nutritional quality can be translated food wise into consuming fish at least twice a week, roots/tuber or cereals, pulses and legumes once a week, fats and oils or sugar and sugar products once a week and vegetables or fruits twice a week. Similar improvements were noticed in other studies but not of similar magnitudes [45,46].

There are several pathways that have been identified through which agricultural interventions influence the nutrition of a household [43–45]. The major pathway through which fish farming contributes to poverty alleviation and economic development in Ghana is the multiplier effect [20]. However, from our findings, the effect here is direct consumption. We conclude, therefore, that there is an increased probability of a fish farming household attaining a higher nutritional quality because of

the ease of access and availability to fish and other nutritious food through direct consumption of their catch and the increase in their purchasing power from the sale of their fish.

**Table 4.** Impact of fish farming participation on household food security.

| Variables | Matching Algorithm | FFHH | nFFHH | ATT | BSE | T-Stat | FFHH | nFHHH |
|-----------|-------------------|------|-------|-----|-----|--------|------|-------|
| FCS | NNM | | | | | | | |
| | (1) | 69.77 | 54.23 | 15.54 | 1.71 | 9.11 | 143 | 3867 |
| | NNM | | | | | | | |
| | (5) | 69.77 | 54.28 | 15.54 | 1.44 | 10.78 | 143 | 3867 |
| | KBM (0.03) | 69.77 | 55.30 | 13.86 | 1.38 | 10.31 | 143 | 3867 |
| | KBM (0.06) | 69.50 | 55.64 | 13.86 | 1.34 | 10.35 | 143 | 3867 |

BSE = Bootstrapped standard errors with 100 replications. NNM: Nearest Neighbor matching, KBM: Kernel-based matching, ATT: Average Treatment on the Treated, t-stat: T statistic, FFHH: Fish farming household, nFFHH: non-fish farming households.

## 4.4. Test for Selection Bias

The results to check the matching procedure are shown in Table 5. The results show a significant percentage reduction in biases as seen in Table 5 and most importantly, after matching, no significant differences are seen between fish farming and non-fish farming households for any of the covariates.

**Table 5.** Tests for selection bias after matching.

| Variables | Matched Sample | | | | |
|-----------|------|-------|-------|--------------|-----------------|
| | FFHH | nFFHH | %Bias | Bias Reduced | T-Test *p*-Value |
| Wealth index squared | 2.16 | 1.86 | 10.8 | 47.4 | 0.48 |
| Wealth index | 0.12 | 0.11 | 1.0 | 95.8 | 0.94 |
| Education | 7.76 | 6.76 | 15.7 | 47.4 | 0.18 |
| Age | 45.99 | 46.24 | −2.1 | 82.8 | 0.88 |
| Peri-urban | 0.21 | 0.24 | −8.2 | 79.5 | 0.57 |
| Married | 0.93 | 0.92 | 2.1 | 94.2 | 0.82 |
| Employed | 0.97 | 0.99 | −10.3 | 22.9 | 0.41 |
| Sex | 0.92 | 0.93 | −2.2 | 91.5 | 0.82 |
| Ecology | 2.62 | 2.57 | 7.8 | 80.4 | 0.50 |
| Income/capita | 63.09 | 56.01 | 1.8 | 96.2 | 0.61 |
| Household size | 7.69 | 8.36 | −16.3 | 68.8 | 0.24 |

FFHH: Fish farming household, nFFHH: non-fish farming households.

After the matching procedure, the pseudo-$R^2$ and the likelihood ratio reduced in magnitude and were insignificant (Table 6). This finding supports the point that there are no significant differences in terms of the relevant covariates between a fish farming and a non-fish farming household.

**Table 6.** Statistical tests to evaluate the matching procedure.

| Matching Algorithm | $R^2$-Before | $R^2$-After | LR-Before | LR After | Chi$^2$-Before | Chi$^2$-After | MSB-Before | MSB-After |
|--------------------|--------------|-------------|-----------|----------|----------------|---------------|------------|-----------|
| NNM (1) | 0.15 | 0.01 | 189.05 | 4.54 | 0.00 | 0.95 | 30.90 | 7.10 |
| NNM (5) | 0.15 | 0.01 | 189.05 | 5.50 | 0.00 | 0.99 | 30.90 | 5.50 |
| KBM (0.03) | 0.15 | 0.01 | 189.05 | 3.22 | 0.00 | 0.99 | 30.90 | 3.50 |
| KBM (0.06) | 0.15 | 0.03 | 189.05 | 10.79 | 0.00 | 0.46 | 30.90 | 8.10 |

MSB: Mean squared bias. LR: Likelihood ratio, Chi$^2$: Chi-squared.

The test of joint significant effect of the covariates was rejected post matching with an insignificant likelihood ratio in all four different versions of the nearest neighbor and kernel-based approaches. The mean bias reductions according to the literature is acceptable when it is below 20 percent. From Table 6, all our reductions are below 11 percent. This is an indication of a good matching procedure.

### 4.5. Testing for Hidden Bias Post Estimation

The presence of bias from observed covariates was tested using the Rosenbaum sensitivity analysis. Under the assumption that the true treatment effect was underestimated (sig−), the bounds show that the results are highly unresponsive to the presence of hidden bias. Results are in Table 7. The sensitivity analysis results imply that our results, which say that adoption of fish farming has a positive correlation with nutritional quality, are robust. The sensitivity analysis results also imply that if a fish farming program is implemented by the government or an entity, we expect that the impact on household nutritional quality should be at least 13.9 upwards.

**Table 7.** Sensitivity analysis with Rosenbaum Bounds.

| Gamma | Sig+ | Sig− |
|:---:|:---:|:---:|
| 1 | $4.1 \times 10^{-14}$ | $4.1 \times 10^{-14}$ |
| 1.1 | $1.4 \times 10^{-12}$ | $7.8 \times 10^{-16}$ |
| 1.2 | $2.8 \times 10^{-11}$ | 0 |
| 1.3 | $3.5 \times 10^{-10}$ | 0 |
| 1.4 | $2.9 \times 10^{-9}$ | 0 |
| 1.5 | $1.8 \times 10^{-8}$ | 0 |
| 1.6 | $9.2 \times 10^{-8}$ | 0 |
| 1.7 | $3.7 \times 10^{-7}$ | 0 |
| 1.8 | $1.3 \times 10^{-6}$ | 0 |
| 1.9 | $3.9 \times 10^{-6}$ | 0 |
| 2.0 | 0.00001 | 0 |

## 5. Post-Estimation Analysis

### 5.1. Effects of Ecological Zone and Female-Headed Households on Nutritional Quality

Our results corroborate the World Food Program's report, which says that the Upper East, North and Upper West regions have the highest number of households that are either severely or moderately food insecure [47]. From our results, we find that ecological location of the household is a significant determinant in adopting fish farming and the probability increases as one moves inland. The fishery sector in Ghana is 40 percent women [48]. The role of women in aquaculture in Ghana includes setting prices, marketing and processing. [49] Women in Northern Ghana do not get the same opportunities in terms of land and labor for farming or income generating opportunities compared to men in the same community. This contributes to them falling into a cycle of food insecurity. [25,37].

Based on our findings, we conducted a post-estimation analysis in order to inform a policy geared towards improving household food and nutritional security status. We use the FCS thresholds: poor (0–21), borderline (21.5–35) and acceptable (>35) as a dependent variable and calculate the probabilities of households belonging to any of these categories. Most of the sample households were in the rural area, so we are interested in the best way to help rural households improve their food security status in a cost-effective way. The variables of interest we analysed were being a fish farmer, living in a household with an educated female head and living in the rural savannah ecological zone. The interest in the savannah ecological zone is because the three Northern regions (Upper East, Upper West and Northern) have the highest prevalence of food insecurity (Table 8). In addition, from our analysis, we observed that moving away from the coast towards inland (forest and savannah ecological zones) increases the probability of the adoption of cane/pen fish farming.

**Table 8.** Food Consumption scores by region.

| Region | Min (FCS) | Max (FCS) |
|---|---|---|
| Western | 32.00 | 84.50 |
| Central | 39.00 | 88.00 |
| Greater Accra | 60.00 | 88.00 |
| Volta | 31.50 | 89.50 |
| Eastern | 34.50 | 85.00 |
| Ashanti | 25.00 | 112.02 |
| Brong Ahafo | 17.30 | 109.00 |
| Northern | 33.00 | 71.00 |
| Upper East | 30.50 | 73.50 |
| Upper West | 27.00 | 73.50 |

Based on our findings, we conducted a post-estimation analysis in order to inform policy geared towards improving household food and nutritional security status. We use the FCS thresholds: poor (0–21), borderline (21–35) and acceptable (>35) as a dependent variable and calculate the probabilities of households belonging to any of these categories. Most of the sample households were in the rural area, so we are interested in the best way to help rural households improve their food security status in a cost-effective way. The variables of interest we analysed were being a fish farmer, living in a household with an educated female head and living in the rural savannah ecological zone. The interest in the savannah ecological zone is because the three Northern regions (Upper East, Upper West and Northern) have the highest prevalence of food insecurity (Table 8). In addition, from our analysis, we observed that moving away from the coast towards inland (forest and savannah ecological zones) increases the probability of adoption of cane/pen fish farming.

The post estimation results show that a household in the rural savannah ecological zone with a female household head engaged in fish farming, has a higher probability of being food secure (96 percent), as shown in Table 9.

**Table 9.** Probability of improving food security in the rural savannah zone if household head is a female fish farmer.

| Variable | Predicted Prob. |
|---|---|
| poor | 0.001 * |
| | (0.00) |
| borderline | 0.036 * |
| | (0.01) |
| acceptable | 0.963 *** |
| | (0.01) |
| Observations | 4000 |

Standard errors in parentheses. * $p < 0.1$, ** $p < 0.05$, *** $p < 0.01$ NOTE: ALL other regressors at their mean value except for FFHH = 1, Peri-urban = 0, SexHH = 0 (female) and ecology = 3 (savannah zone).

### 5.2. Policy Recommendations

The first policy suggestion from our findings is the promotion of fish farming in the three Northern regions, preferably aquaculture in water reservoirs using cages or pens. The Northern regions are the least developed in the country and so require a fish farming system that has a high rate of return as the fish reach market size faster [49]. Bamboo, PVC pipes, drums and cane are materials that can be used to construct the cage as well as intensive feeding with commercial feeds. [40]. Small to medium scale cage aquaculture is less costly compared to land-based ponds because of relatively lower initial capital requirements and is easier to manage and monitor production [40,49].

The existing irrigation schemes in the North and Upper East regions (Bontanga, Golinga, Ligba, Vea and Tono) for rice and vegetable production will be good sources of water for the cage fish farms.

The second policy suggestion is to encourage women to engage in more than the processing of fish and getting involved in production as well. Our post estimation has positive implications for increasing household nutritional quality with women as heads of households. As the source of income in the household is diversified, the probability of being food insecure reduces. Homemakers can support their husbands and contribute to the household income by adopting fish farming on a small scale on a piece of land at home or close to the home.

Finally, a study that uses a repeated cross section approach to assess the impact fish farming on household nutritional quality over a period is suggested.

## 6. Conclusions

The study contributes to the literature by evaluating the impact of participating in fish farming on the nutritional quality of households in Ghana. We evaluated both direct impact pathways (consumption of fish) and indirect impact pathways (sale of fish to buy other foods) in this study using the Food Consumption Score (FCS) as a proxy measure for food security.

On the margin, the probability of adopting fish farming increased with wealth, ecological zone, being a resident in a peri-urban area and household size while it declined with per capita household income. The average treatment effect of adopting fish farming on the food security of fish farming households showed an increase in the range of 13.9 to 15.5 points. This translates into consuming fish twice in a week, roots/tuber or cereals once a week, pulses and legumes once a week, fats and oils or sugar and sugar products once a week and vegetables or fruits twice a week. We infer that fish farming increases the diversity and frequency of food consumed through direct consumption and not so much through the income effect.

Most importantly, our analysis showed that households in the savannah zone have an opportunity to engage in fish farming; especially those in the rural areas have a higher probability of improving their food security status. The provision of institutional, financial and technical support such as access to land, supply of seeds and feed, water resources, extension services and market services particularly for low-income households and women to engage in fish farming would help to improve food security. This would be especially true in Northern Ghana where poverty and malnutrition levels are highest.

**Supplementary Materials:** The following are available online at http://www.mdpi.com/2071-1050/11/10/2807/s1.

**Author Contributions:** Conceptualization, K.Q. and A.A.; Methodology, X.X.; Formal Analysis, A.A.; Investigation, A.A.; Data Curation, A.A.; Writing—Original Draft Preparation, A.A.; Writing—Review & Editing, A.A. and K.Q.; Funding Acquisition, K.Q.

**Funding:** This research was supported by funding from the United States Agency for International Development (USAID); Cooperative Agreement No. *EPP-A-00-06-00012-00* and by contributions from participating institutions.

**Acknowledgments:** This project was supported by the Feed the Future Innovation Lab for Collaborative Research on Aquaculture & Fisheries through the United States Agency for International Development (USAID); Cooperative Agreement and by contributions from participating institutions. The Faculty of Renewable Natural Resources, Kwame Nkrumah University of Science, provided logistics and Technology (KNUST), a partner institution, students and professors from the Faculty of Food Science, KNUST in Ghana.

**Conflicts of Interest:** The authors confirm that there are no known conflicts of interest associated with this publication and there has been no significant financial support for this work that could have influenced its outcome.

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
