# Peer review of "Assessment of Household Food Security in Fish Farming Communities in Ghana"

_sustainability, doi:10.3390/su11102807_

Round 1

Reviewer 1 Report

Comments and suggestions are highlighted in yellow on the manuscript (attached)

Author Response

Reviewer 1 Comments Responses

The sentences that had been highlighted and asked to be revised for clarity have been revised accordingly.

Lines 295, 296: The Issue of access and availability being achieved through multiplier effects have been clarified to incorporate the fact that multiplier effects form the sale of seafood for cash increases the purchasing power of households which gives them access to other types f seafood and food sources in general.

Line 316: The results of the Rosenbaum Bias test implies that our results are not affected by hidden bias from confounding variables that were not measured in the data. The Rosenbaum Sensitivity Analysis tells us that the results we got are an underestimation of the effect of fish farming on nutritional quality. So that with the addition of variables and assumptions mimicking more real conditions, we should get a higher effect (> 15.5). The policy implication here is that in the implementation of a fish farming program in rural areas, we should see an increase in household nutritional quality.

Line 325: More information has been provided on why female-headed households are food insecure. Women in agriculture are not privy to the same resources and income-earning opportunities that their male counterparts are able to obtain and partake in.

Line 337: adoption of cane/pen fish farming.

Reviewer 2 Report

The MS is a study about effects of aquaculture in Ghana on food consumption and wealth on household level. Further, the study investigates factors that are increasing the adoption of aquaculture.

The topic of the study is relevance. However, I have some general and some specific remarks, which should be taken into account before publishing.

General: The study is not discussing critically the implementation of aquaculture. As this journal is named “Sustainability”, I would expect that aquaculture is more critically discussed at least in the three dimensions of social, environmental and economic sustainability. Surely, aquaculture offers possibilities but there are also risks. So, the suitability of specific aquaculture technologies need to discussed depending on the local circumstances including the risks.

Do farmers get subsidies for running aquaculture (or is the sector subsidized)?

The MS is not mentioning the access in Ghana (and the single regions) to major aquaculture resources: Feed (which are making often 70% of the entire production cost); water, fingerlings; and markets.

Further, the MS should more in detail describe what the difference is between the different applied aquaculture systems: Pond aquaculture is often run as polyculture; depending on the intensity, the feed can range from natural pond products/kitchen/farm by-products and the fish might reproduce in the pond. Further, having a pond is ensuring water access for other activities (vegetables/animals) which are increasing the on-farm diversification. So, improvement in food consumption can already be measured by a more constant water access. Cage culture is depending completely on pellet feed and fingerling from external sources; so it requires financial input and is not often suitable for poor farmers.

The study showed that wealthier households are more likely to implement aquaculture, but it was also shown, that households which are running aquaculture are having a substantial lower income. How does this fit together? Please discuss this point.

Line 21: Please write what this number (15.5) is (e.g. food consumption score) or do not mention the number here. Further, this number is misleading as the averages lay between 13.9 and 15.5, see line 283, (and is not 15.5).

Line 147: The “non-fish farming households”: did these farmers actively decide not to invest in aquaculture or were these farmers never having the choice to implement aquaculture? Were these farmers having the same access to the essential resources like water, feed, markets and knowledge?

Line 157: Can you please explain more in detail how you got the data from/about farmers about what “would have happened”?

Line 158: What does the “F” mean?

Table 2: Please describe the rows more clearly (or explain the abbreviations below the table). Please also include the units in the rows.

Line 217: Please discuss intensively, that fish farmers are wealthier but have a substantially lower income.

Table 3: Why are some rows bold?  If it should mark the significant effects than the substantially reduced income should be bold, too.

Line 277: Obviously, in this chapter you went through all the significant effect. But you were missing the reduced income of aquaculture farmers. Please include it.

Table 4 & 6: Please explain this table more in detail or explain the abbreviations.

Line 324: Please go a bit into detail and describe the role of female in the fish sector. Women are often engaged in the processing and marketing. Can you give numbers how often women are household heads and are running aquaculture?

Line 348: I disagree that cage culture is a low cost activity. These activities require costly external feeds and fingerlings. The reference [39] seems to me not sound enough alone to generalize this statement (from aquaculture perspective).  

Line 264: I am missing in the MS a more detailed description about fish as part of the nutrition as results of your study. Only one sentence (line 284) described it a bit. Please further describe the general importance of fish in Ghana and also, whether the fish species produced in aquaculture are excepted by the consumers.

Line 369: in line 139 you said it was between 13.9 and 15.5.

Line 379: I disagree with this last sentence.

Author Response

Reviewer 2 Comments

The MS is a study about effects of aquaculture in Ghana on food consumption and wealth on household level. Further, the study investigates factors that are increasing the adoption of aquaculture.

The topic of the study is relevance. However, I have some general and some specific remarks, which should be taken into account before publishing.

1.       General: The study is not discussing critically the implementation of aquaculture. As this journal is named “Sustainability”, I would expect that aquaculture is more critically discussed at least in the three dimensions of social, environmental and economic sustainability. Surely, aquaculture offers possibilities but there are also risks. So, the suitability of specific aquaculture technologies need to discussed depending on the local circumstances including the risks.

Ans: We have provided some text in the Introduction on the economic, social and environmental sustainability of aquaculture in Ghana.

2.       Do farmers get subsidies for running aquaculture (or is the sector subsidized)?

Ans: No, fish farmers do not get any direct subsidies, and the sector is not subsidized. However, there is a ban on importation of tilapia, the main aquaculture fish species.

3.       The MS is not mentioning the access in Ghana (and the single regions) to major aquaculture resources: Feed (which are making often 70% of the entire production cost); water, fingerlings; and markets.

Ans: The focus of this study is on the demand side so we tried to exclude fish production issues such as feed cost and technology.

4.       Further, the MS should more in detail describe what the difference is between the different applied aquaculture systems: Pond aquaculture is often run as polyculture; depending on the intensity, the feed can range from natural pond products/kitchen/farm by-products and the fish might reproduce in the pond. Further, having a pond is ensuring water access for other activities (vegetables/animals) which are increasing the on-farm diversification. So, improvement in food consumption can already be measured by a more constant water access. Cage culture is depending completely on pellet feed and fingerling from external sources; so, it requires financial input and is not often suitable for poor farmers.

Ans: Again, the focus of this study is on the demand side so we tried to exclude fish production issues such as feed cost and technology.

5.       The study showed that wealthier households are more likely to implement aquaculture, but it was also shown, that households which are running aquaculture are having a substantially lower income. How does this fit together? Please discuss this point.

Ans: Wealth in the study is not measured in monetary value but using the wealth index which is constructed using assets that are owned by the household. Wealth represents a more stable economic standing of the household compared to income or consumption. The index was created using the Principal Components Analysis (PCA) recommended by Filmer and Pritchett (2001). We realized that most of the fish farming households that we surveyed were not earning enough income and so opted to become fish farming in the hope of improving their income-earning opportunities.

6.       Line 21: Please write what this number (15.5) is (e.g. food consumption score) or do not mention the number here. Further, this number is misleading as the averages lay between 13.9 and 15.5, see line 283, (and is not 15.5).

Ans: The Food Consumption score has been removed from the abstract and the corrections and clarification have been made in the manuscript.

7.       Line 147: The “non-fish farming households”: did these farmers actively decide not to invest in aquaculture or were these farmers never having the choice to implement aquaculture? Were these farmers having the same access to the essential resources like water, feed, markets, and knowledge?

Ans: The non-fish farming households from the survey and discussions we had with them were not participants because they were not aware of such a program available in their community and some others were not fully educated about fish farming and its advantages so decided not to take the risk of adopting fish farming.

8.       Line 157: Can you please explain more in detail how you got the data from/about farmers about what “would have happened”?

Ans: The counterfactual data is obtained by creating a similar-statistically looking group through the propensity score matching approach. Propensity score matching allows a researcher to create a control group where you can match the

9.       Line 158: What does the “F” mean?

Ans: The F denotes the participation in fish farming which is fully explained in the supplementary material. We have removed that sentence form the manuscript since it brings up some confusion.

10.   Table 2: Please describe the rows more clearly (or explain the abbreviations below the table). Please also include the units in the rows.

Ans: The variables have been fully explained below the table and the units have been provided in the second column.

11.   Line 217: Please discuss intensively, that fish farmers are wealthier but have a substantially lower income.

Ans: We realized that most of the fish farming households that we surveyed were not earning enough income and so opted to become fish farming in the hope of improving their income-earning opportunities. In short, most of the participants were secondary fish farmers, meaning that they are fish farming to improve their income-earning opportunities. 

12.   Table 3: Why are some rows bold?  If it should mark the significant effects than the substantially reduced income should be bold, too.

Ans: The bold rows show the significant variables, so the reduced income has been made bold too. Thank you.

13.   Line 277: Obviously, in this chapter, you went through all the significant effect. But you were missing the reduced income of aquaculture farmers. Please include it.

Ans: The reduced income variable is significant but since its effect is zero. And we are more interested in the variable that has some a significant impact with value, as such we decided not to explain it in the discussion.

14.   Table 4 & 6: Please explain this table more in detail or explain the abbreviations.

Ans: The abbreviations have been explained below the tables.

15.   Line 324: Please go a bit into detail and describe the role of females in the fish sector. Women are often engaged in processing and marketing. Can you give numbers how often women are household heads and are running aquaculture?

Ans: The role of women in the fishing sector in Ghana ranges from setting prices at the landing site, processing, and marketing. Women do not own boats or canoes and even when they sell the fish, they do not control the money. W did not find data on female household heads who are involved in aquaculture.

16.   Line 348: I disagree that cage culture is a low-cost activity. These activities require costly external feeds and fingerlings. The reference [39] seems to me not sound enough alone to generalize this statement (from aquaculture perspective). 

Ans: You are right that cage culture is expensive. The statement and conclusion have been revised to reflect a more cost-effective system for Northern Ghana.

17.   Line 264: I am missing in the MS a more detailed description about fish as part of the nutrition as results of your study. Only one sentence (line 284) described it a bit. Please further describe the general importance of fish in Ghana and, whether the fish species produced in aquaculture are excepted by the consumers.

Ans: Fish makes up about 60% of the animal protein consumed by Ghanaian households. The commonest species preferred and farmed in Ghana are Tilapia and catfish. This information has been added to the manuscript.

18.   Line 369: in line 139 you said it was between 13.9 and 15.5.

Ans: Corrected to reflect that the same range as mentioned earlier on in the manuscript.

19.   Line 379: I disagree with this last sentence.

Ans: The sentence has been revised.

Round 2

Reviewer 2 Report

The MS got improved by following most of the reviewers’ comments. However, before publications some corrections are needed.

In general:

-          Still I am recommending to include the challenges of the feed supply for the aquaculture sector and the different production systems (pond and cage) into the discussion of the MS. In your responses to the reviewers you said that you exclude these factors. But if you are writing recommendations in the end of the MS and exclude the important factors of feed availability and do not differentiate in your recommendations between the two production systems, the recommendations are knowingly misleading.

-          I still expect to more deeply include into the MS the fact of reduced income generation by aquaculture farmers. In the response to reviewers you state that you exclude this result from discussion as in your analysis this parameter had no effect. But in other sections you state, the income generation was an important argument to actually implement aquaculture.  

Line 66: “…in Ghana can develop towards a sustainable ….”

Line 70-71: Here you state that the most consumed fish are tilapia and catfish. According to FAO data that I read, more than 75% of the aquatic food consumed in Ghana are pelagic fish which are always coming from fisheries and not aquaculture. It will be true that tilapia and catfish are the most consumed aquaculture products. But fishery products seem to be the main source of aquatic food in Ghana.   

Line 78: Here you state that food insecurity is caused by irregularities in season and production mainly caused by irregularities in the rainfall. As aquaculture is also heavily depending on water availability, I would expect statements that these irregularities in the rainfall are also negatively affecting the implementation of aquaculture and result in high risks for aquaculture producers to lose their fish stocks.

Line 127: “Households will further be able to …”

Line 134: Here you state the word “fisheries”; is aquaculture included in this word?

Table 2: Please include the N for the fish-farming HHs and non-fish-farming HHs. Here you give the means and the SDs but no N; I assume that the N for fish-farming HHs is 144 but by looking at the two means/SDs and the pooled data, I am not sure what the N of the non-fish-farming HHs might be.

Line 266: Can you simply add the wealth index squared (1.2%) and the wealth index (2.9%) to sum it it up to 4.1%?

Line 280-281 and 296: How do you explain these contradictions to other reports?

Line 292: Here you state “minimum investment”. Please discuss this more in detail and include the needed investments in feeds/nets/seeds.

Line 319-320: Please replace seafood with fish as freshwater aquaculture does not produce seafood.

Table 5: Please harmonize with table 2 and replace “treated” with fish farming HHs and “control” with non-fish farming HHs.

Line 386-388: Please include here that intensive feeding of costly external feeds is needed.

Line 393 ff: To enable females to run aquaculture, especially the social surrounding (land ownerships, credits…) need to change. This is a long term challenge for the entire society which needs to be enabled by all levels in the society and policy.

Line 414: Here the inclusion of the needed supply with feeds and seeds fits in.

Line 418: “…culture in Northern Ghana in existing water bodies.” I would delete the last part of the sentence as you give no proof for the statements that cage culture is more profitable than pond aquaculture.  

Author Response

In general:

Still I am recommending to include the challenges of the feed supply for the aquaculture sector and the different production systems (pond and cage) into the discussion of the MS. In your responses to the reviewers you said that you exclude these factors. But if you are writing recommendations in the end of the MS and exclude the important factors of feed availability and do not differentiate in your recommendations between the two production systems, the recommendations are knowingly misleading.

Ans: We have included some discussion on production systems and feed in the Introduction.

I still expect to more deeply include into the MS the fact of reduced income generation by aquaculture farmers. In the response to reviewers you state that you exclude this result from discussion as in your analysis this parameter had no effect. But in other sections you state, the income generation was an important argument to actually implement aquaculture. 

Ans: Though the reasons given to us during our survey for this study showed that most fish farmers went into the occupation because of income earning potential, we did not find that in our analysis.

Line 66: “…in Ghana can develop towards a sustainable ….”

Ans: Thank you for the suggestion, the suitable change has been made.

Line 70-71: Here you state that the most consumed fish are tilapia and catfish. According to FAO data that I read, more than 75% of the aquatic food consumed in Ghana are pelagic fish which are always coming from fisheries and not aquaculture. It will be true that tilapia and catfish are the most consumed aquaculture products. But fishery products seem to be the main source of aquatic food in Ghana.  

Ans: We have revised the statement.

Line 78: Here you state that food insecurity is caused by irregularities in season and production mainly caused by irregularities in the rainfall. As aquaculture is also heavily depending on water availability, I would expect statements that these irregularities in the rainfall are also negatively affecting the implementation of aquaculture and result in high risks for aquaculture producers to lose their fish stocks.

Ans: It is true that irregular rainfall affects aquaculture, however the common practice in Ghana is that farmers commonly situate their farms close to water bodies. Some also dig bore holes to have an alternate source of water especially in the dry season.

Line 127: “Households will further be able to …”

Ans: Thank you for the suggestion, the suitable change has been made.

Line 134: Here you state the word “fisheries”; is aquaculture included in this word?

Ans: Yes, aquaculture has been added to this sentence.

Table 2: Please include the N for the fish-farming HHs and non-fish-farming HHs. Here you give the means and the SDs but no N; I assume that the N for fish-farming HHs is 144 but by looking at the two means/SDs and the pooled data, I am not sure what the N of the non-fish-farming HHs might be.

Ans: The N for the non-fish farming households used is 300. The reason for the slightly higher number is to make sure that in the matching process, each fish farming household is matched under both matching algorithms.

Line 266: Can you simply add the wealth index squared (1.2%) and the wealth index (2.9%) to sum it up to 4.1%?

Ans: We cannot simply sum up wealth index and wealth index squared as they represent different econometric relationship between the wealth of the household and their nutritional quality status. Though our interest is in the wealth index squared variable, it is econometrically important to show the linear relationship first before moving on to estimate the non-linear relationship between two variables.

Line 280-281 and 296: How do you explain these contradictions to other reports?

Ans: There are a number of reasons for the contradictions to other reports. One is the ecology of the areas where the data was collected in the different studies. There were originally 10 regions in Ghana (now 16) and no two studies collected data from the same ecological zones. Another reason could be the season when the studies were carried out, and the cocktail of variables used in other studies are not the same combination as in our study.

Line 292: Here you state, “minimum investment”. Please discuss this more in detail and include the needed investments in feeds/nets/seeds.

Ans: The minimum investment here is in terms of pond construction and other labor requirements. We have provided more discussion in the manuscript.

Line 319-320: Please replace seafood with fish as freshwater aquaculture does not produce seafood.

Ans: Seafood had been replaced with fish.

Table 5: Please harmonize with table 2 and replace “treated” with fish farming HHs and “control” with non-fish farming HHs.

Ans: Treated has been changed to Fish farming HHs and control has been changed to non-Fish farming HHs.

Line 386-388: Please include here that intensive feeding of costly external feeds is needed.

Ans: This suggestion has been included in the sentence.

Line 393 ff: To enable females to run aquaculture, especially the social surrounding (land ownerships, credits…) need to change. This is a long-term challenge for the entire society which needs to be enabled by all levels in the society and policy.

Ans: This recommendation is a long-term challenge which was suggested to the government. Other donor agencies have actually picked up on it and we hope that females will be able to participate more than just the processing and marketing of fish.

Line 414: Here the inclusion of the needed supply with feeds and seeds fits in.

Ans: The suggestion was included in the sentence.

Line 418: “…culture in Northern Ghana in existing water bodies.” I would delete the last part of the sentence as you give no proof for the statements that cage culture is more profitable than pond aquaculture. 

Ans: The sentence has been deleted as suggested.